# The translational landscape of the splicing factor SRSF1 and its role in mitosis

**Magdalena M Maslon[1][†], Sara R Heras[1,2][†], Nicolas Bellora[3][‡], Eduardo Eyras[3,4], Javier F Cáceres[1]***

[1]MRC Human Genetics Unit, Institute of Genetics and Molecular Medicine, University of Edinburgh, Edinburgh, United Kingdom; [2]GENYO, Centre for Genomics and Oncological Research, Pfizer/University of Granada/Andalusian Regional Government, Granada, Spain; [3]Computational Genomics Group, Universitat Pompeu Fabra, Barcelona, Spain; [4]Catalan Institution for Research and Advanced Studies (ICREA), Barcelona, Spain

**Abstract** The shuttling serine/arginine rich (SR) protein SRSF1 (previously known as SF2/ASF) is a splicing regulator that also activates translation in the cytoplasm. In order to dissect the gene network that is translationally regulated by SRSF1, we performed a high-throughput deep sequencing analysis of polysomal fractions in cells overexpressing SRSF1. We identified approximately 1500 mRNAs that are translational targets of SRSF1. These include mRNAs encoding proteins involved in cell cycle regulation, such as spindle, kinetochore, and M phase proteins, which are essential for accurate chromosome segregation. Indeed, we show that translational activity of SRSF1 is required for normal mitotic progression. Furthermore, we found that mRNAs that display alternative splicing changes upon SRSF1 overexpression are also its translational targets, strongly suggesting that SRSF1 couples pre-mRNA splicing and translation. These data provide insights on the complex role of SRSF1 in the control of gene expression at multiple levels and its implications in cancer.

***For correspondence:** Javier.Caceres@igmm.ed.ac.uk

[†]These authors contributed equally to this work

**Present address:** [‡]Laboratorio de Microbiología Aplicada y Biotecnología, Centro Regional Universitario Bariloche, Universidad Nacional del Comahue, INIBIOMA (CONICET-UNComa), Bariloche, Argentina

**Competing interests:** The authors declare that no competing interests exist.

## Introduction

Alternative splicing is a central mechanism for the regulation of gene expression allowing increased proteomic complexity in higher eukaryotes (*Smith and Valcárcel, 2000*; *Braunschweig et al., 2013*; *Kornblihtt et al., 2013*). It is regulated at many different levels, mainly by the binding of protein factors to enhancers and silencers in the pre-mRNA. The importance of chromatin structure and histone modifications in alternative splicing regulation has only begun to emerge recently (*Schwartz et al., 2009*; *Tilgner et al., 2009*; *Luco et al., 2010, 2011*; *Pradeepa et al., 2012*).

The SR proteins are a well-characterized family of splicing factors with a role in both constitutive and alternative splicing (reviewed by *Lin and Fu, 2007*). They have a modular structure consisting of one or two N-terminal RNA recognition motifs (RRMs), which determine their RNA-binding specificity and a C-terminal domain rich in arginine and serine residues (RS domain) (*Shepard and Hertel, 2009*). An extended family of RS domain-containing proteins present in metazoans, termed SR-like or SR-related proteins, are structurally and functionally distinct from canonical SR proteins and have roles not exclusively related to splicing but participate in other cellular functions as well, including transcription and cell cycle progression (*Boucher et al., 2001*). The activity of SR proteins in alternative splicing is defined by the location of their binding sites, generally displaying a stimulatory role in splicing when bound to exons and an inhibitory role when bound to introns (*Han et al., 2011*; *Erkelenz et al., 2013*; *Zhou and Fu, 2013*). Their function in alternative splicing can be antagonized by the activity of hnRNP A/B proteins in a concentration-dependent manner, in such a way that the relative ratios of these antagonists

**eLife digest** Genes contain the instructions to make proteins. These instructions are first transcribed to produce an intermediate molecule called a messenger RNA (mRNA), which is then translated to produce the protein. However, gene sequences are often interrupted by 'introns', sections of DNA that do not code for protein, and these introns must be removed from the mRNA molecules via a process called 'splicing' before the protein is produced.

Splicing can also be used to 'mix and match' sections of gene sequences to produce slightly different versions of the same protein in a process called 'alternative splicing'. SRSF1 is one of a family of proteins that control both types of gene splicing but also promotes the translation of specific mRNAs. To date only a few of the genes whose translation is regulated by SRSF1 have been identified.

Here, Maslon, Heras et al. have used human cells that artificially produce more SRSF1 protein than normal to identify those genes whose translation is regulated by SRSF1. Over 1500 'target genes' were found; many of which encoded proteins that are involved in cell division—and cells with less SRSF1 than normal failed to divide properly. Maslon, Heras et al. also found a link between alternative splicing and protein translation: many of the mRNAs that were spliced differently in cells that over-produced SRSF1 were also genes whose translation was affected by SRSF1.

Since uncontrolled cell division, or defects in mRNA splicing or protein synthesis are all often linked to cancer, these discoveries might provide new insights into the mechanisms underlying this disease.

can influence patterns of regulated splicing in a tissue-specific or developmentally regulated manner (*Eperon et al., 2000*; *Zhu et al., 2001*). Although initially the RS domain was proposed to solely act as a protein–protein interaction domain, it was later revealed that it also acts to contact the pre-mRNA (*Shen and Green, 2004*; *Shen et al., 2004*; *Hertel and Graveley, 2005*). The RS domain has also been shown to determine the localization and nucleo-cytoplasmic shuttling properties of SR proteins (*Cáceres et al., 1997*, *1998*; *Allemand et al., 2001*).

A role for SR proteins and their natural antagonists, hnRNP proteins, in deregulated alternative splicing during cancer progression has been extensively documented (reviewed by *Venables, 2004* and *David and Manley, 2010*). For instance, three hnRNP proteins, hnRNP A1, hnRNP A2 and PTB, control the alternative splicing of pyruvate kinase (PK-M) pre-mRNA giving rise to an isoform that is required for aerobic glycolysis used by rapidly growing tumor cells (*Clower et al., 2010*; *David et al., 2010*; *Chen et al., 2012*). The SR protein SRSF3 (previously known as SRp20) antagonizes the function of the reported hnRNP proteins in PK-M alternative splicing (*Wang et al., 2012*). The antagonistic activities of SRSF1 and hnRNP A1 also control the epithelial-to-mesenchymal transition (EMT) and its reversal (MET) through production of two different alternatively spliced isoforms of the Ron proto-oncogene (*Ghigna et al., 2005*; *Bonomi et al., 2013*). The levels of SRSF1 are regulated during EMT/MET via alternative splicing associated with the nonsense-mediated mRNA decay pathway (AS-NMD), which is regulated by the splicing factor Sam68 (*Valacca et al., 2010*). SRSF1 has been identified as an oncogenic protein with altered expression in several tumors (*Karni et al., 2007*). Its increased expression leads, in cooperation with MYC, to the transformation of mammary epithelial cells (*Anczuków et al., 2012*). SRSF3 has also been proposed to be a proto-oncogene critical for cell proliferation and tumor induction and maintenance (*Jia et al., 2010*), whereas SRSF6 (SRp55) is amplified and is an oncoprotein in lung and colon cancers (*Cohen-Eliav et al., 2013*). Recently, a cellular defense mechanism to deal with the oncogenic potential of increased SRSF1 expression has been described, whereby SRSF1 stabilizes the tumor suppressor protein p53 by blocking its MDM2-dependent proteasomal degradation, which ultimately leads to oncogene-induced senescence (OIS) (*Fregoso et al., 2013*). Interestingly, other splicing factors, such as hnRNP A2/B1, are also overexpressed in some types of cancers, such as glioblastomas, where they are correlated with poor prognosis (*Golan-Gerstl et al., 2011*).

A subset of the SR protein family members shuttle from the nucleus to the cytoplasm, including SRSF1 (SF2/ASF), SRSF3 (SRp20), SRSF4 (SRp75), SRSF6 (SRp55), SRSF7 (9G8), and SRSF10 (SRp38)

(*Cáceres et al., 1998*; *Cowper et al., 2001*; *Cazalla et al., 2002*; *Sapra et al., 2009*). Importantly, shuttling SR proteins have been shown to participate in a wide range of post-splicing activities, including mRNA nuclear export, nonsense-mediated mRNA decay, and mRNA translation (reviewed by *Long and Caceres, 2009* and *Twyffels et al., 2011*). As an example, several studies have revealed that three shuttling SR proteins, SRSF1, SRSF3, and SRSF7, can act as mRNA export adaptors via their interaction with the cellular export factor TAP (*Huang and Steitz, 2001*; *Huang et al., 2003*; *Hargous et al., 2006*). Furthermore, increased concentration of SRSF1 promotes nonsense-mediated decay (NMD) (*Zhang and Krainer, 2004*; *Sato et al., 2008*). A number of SR protein family members were found to have a role in translation. We have previously shown that hypophosphorylated SRSF1 protein is associated with polyribosomes in cytoplasmic extracts and enhances translation in HeLa cells both in vitro and in vivo (*Sanford et al., 2004*, *2005*). Furthermore, we also uncovered the molecular mechanism by which SRSF1 promotes translation by showing that it promotes translation initiation of bound mRNAs by suppressing the activity of 4E-BP, a competitive inhibitor of cap-dependent translation. This activity is mediated by interactions of SRSF1 with components of the mTOR signaling pathway (*Michlewski et al., 2008*). In agreement with this, it was also shown that SRSF1 activates the mTORC1 branch of the pathway, as measured by S6K and 4E-BP1 phosphorylation (*Karni et al., 2008*). These findings suggest a model whereby SRSF1 acts as an adaptor protein that recruits the signaling molecules responsible for regulation of cap-dependent translation of specific mRNAs. Another shuttling SR protein, SRSF7, has also been shown to promote translation of unspliced MMPV retroviral transcripts (*Swartz et al., 2007*), whereas SRSF5 and SRFS6 increase the rate of Gag translation in the HIV virus (*Swanson et al., 2010*). SRSF3 functions as a trans-acting factor for the internal ribosome entry site (IRES)-mediated translation of poliovirus, which requires its cytoplasmic relocalization during viral infection (*Bedard et al., 2007*; *Fitzgerald and Semler, 2011*, *2013*). Despite the presence of growing evidence for a role for shuttling SR proteins in the regulation of mRNA translation, only very few physiological targets have been identified. This raises the issue whether this activity of SR proteins has an important role in gene expression and/or whether it is associated with a particular cellular pathway.

Here, we have focused on the identification of the mRNA translational targets of the SRSF1 protein. We carried out high-throughput deep sequencing analysis of polysomal fractions in mammalian cells overexpressing SRSF1. This resulted in the identification of a large number of mRNAs that are transla-tionally regulated by SRSF1. These mRNAs encode proteins involved in cell cycle regulation, such as spindle, kinetochore, and M phase proteins, which are essential for accurate chromosome segregation. Interestingly, we also observed that in many cases SRSF1 affects the alternative splicing of a subset of mRNAs and also influences translation of these isoforms, suggesting a role for SRSF1 in the coupling of pre-mRNA splicing and translation. Altogether, the finding that SRSF1 promotes the increased translation of genes associated with cell division could partially explain the oncogenic role of SRSF1. In summary, these data provide insights on the complex role of SRSF1 in the control of gene expression and its implications in cancer.

## Results

### Identification of SRSF1 translational targets

In order to identify SRSF1 translational mRNA targets, we performed a polysomal shift analysis to follow mRNAs that move from the subpolysomal fraction to the heavier polysomal fractions in HEK 293T cells upon increased SRSF1 expression. Maintaining proper levels of SRSF1 could be critical for cell function. As such, SRSF1 expression is subjected to negative autoregulation in order to maintain homeostatic levels, which involves multiple layers of post-transcriptional and translational control (*Sun et al., 2010*). Thus, in order to avoid cellular mechanisms that could limit an increased SRSF1 expression, we relied on transient overexpression of an epitope-tagged SR protein cDNA encoding wild-type SRSF1 protein. We used two different concentrations of the SRSF1 expression vector and obtained a maximum of a threefold increase in the levels of transfected SRSF1 protein over endogenous protein in HEK 293T cells that displayed approximately 80–90% transfection efficiency (*Figure 1—figure supplement 1*). We chose the highest concentration of transfected SRSF1 protein since this resulted in maximum activation of a luciferase reporter harboring an SRSF1 binding site (*Sanford et al., 2004*) (*Figure 1—figure supplement 2*, left panel). The translational activation of the luciferase reporter induced by SRSF1 correlated well with a threefold increase in the polysomal/subpolysomal ratio of the

reporter RNA (*Figure 1—figure supplement 2*, right panel). The expression of SRSF1 varies widely in a tissue-specific manner and differences of up to 20-fold between different tissues have been reported (*Zahler et al., 1993*; *Hanamura et al., 1998*). Thus, this level of overexpression is within physiological levels and correlates well with the maximum activation of a translational reporter (*Figure 1—figure supplements 1 and 2*). We proceeded to fractionate cell cytoplasm across 10–45% sucrose gradients and isolated RNA from the subpolysomal and heavy polysomal fractions from control cells and from cells transiently overexpressing SRSF1.

Next, we identified by high-throughput sequencing analysis those mRNAs that shifted to the polysomal fractions upon SRSF1 increased expression (*Figure 1A*). It has been shown that calculating mRNA translation levels as log ratios of actively translated mRNAs divided by the corresponding cytoplasmic mRNA results in a significant number of false positives and false negatives (*Larsson et al., 2010*). Therefore, to precisely identify those mRNAs whose translation is responsive to increased levels of SRSF1, we used for normalization the log ratios of polysomal mRNAs versus RNAs in subpolysomal plus polysomal fractions. The resulting polysome index measures the proportion/density of each transcript that is present in the polysomal fractions. We compared empty vector-transfected to SRSF1-transfected cells by calculating the distribution of the $\log_2$ ratios of their respective polysome index, which we defined as the polysome shift ratio (PSR) (*Figure 1B*). This allowed for the scoring of an increase in translational efficiency independently of the cellular abundance of the corresponding transcript. A cut off of 0.889 (p<0.01), corresponding to a 1.85-fold increase in the proportion of polysomal-associated transcripts, resulted in the identification of 1576 mRNAs that shifted to heavier polysomal fractions upon SRSF1 overexpression (*Figure 1B* and data in Supplementary file 1 at Dryad: *Maslon et al. (2014)*). Gene Ontology analyses showed that a large proportion of those mRNAs identified in the polysomal shift analysis encode for proteins involved in cell cycle regulation, mitosis, transcription, and post-translational protein modification (based on analysis with DAVID [*Dennis et al., 2003*]) (*Figure 2—figure supplement 1A*). Among the RNA targets that displayed a polysomal shift are mRNAs encoding proteins related to cancer, such as NRAS, the Ras-related protein R-Ras2, and those related to cell cycle, such as CDC27, the retinoblastoma binding protein RBBP8, and retinoblastoma-like 1 (RBL1). (A list of all SRSF1 translational targets is provided in Supplementary file 1 at Dryad: *Maslon et al. (2014)*.) The SRSF1 translational targets may represent indirect as well as direct events. In the first scenario, increased expression of SRSF1 could lead to general changes in gene expression, which could result indirectly in the translational upregulation of a subset of mRNAs. Conversely, direct events would represent events whereby SRSF1 binds to its mRNA targets and activates their translation. Interestingly, we observed that 41% of all mRNAs identified in the polysomal shift experiment upon SRSF1 overexpression were previously identified as bona fide RNA targets of this SR protein by CLIP-seq (*Sanford et al., 2009*) (*Figure 2—figure supplement 1B*). This strongly suggests that these are direct mRNA translational targets of SRSF1. We combined k-mer enrichment analysis (the enrichment of every 5-mer within the RNA sequences) with motif discovery to search for over-represented sequences in SRSF1 mRNA translational targets. MEME was used to retrieve a motif logo from mRNA regions containing the over-represented 5-mers (*Bailey and Elkan, 1994*, *1995*). This resulted in the identification of a purine-rich motif very similar to the one obtained when identifying genome-wide targets of SRSF1 (*Figure 2A*) (*Sanford et al., 2008*, *2009*). Interestingly, the frequency of this motif showed a clear gradient, being more predominant in CLIP-positive translational targets than in CLIP-negative translational targets and was even more reduced in both the CLIP-positive and CLIP-negative subset of mRNAs that did not shift to polysomes following SRSF1 overexpression (*Figure 2B*). In fact, statistical analysis showed a significant enrichment of the CLIP-positive mRNAs containing the identified consensus motif (CM) in the SRSF1 translational targets (Fisher's exact test: OR 1.686, p<2.2E-16) (*Figure 2C*). We further refer to these 505 mRNAs as SRSF1 direct translational targets. Next we analyzed whether there was any position bias with respect to the SRSF1 consensus motif in SRSF1 translational targets. We observed that this motif is preferentially located in the coding DNA sequence (CDS) and to a lesser extent in the 5′UTR of SRSF1 translational targets, when compared with those mRNAs whose translation is unaffected by increased SRSF1 expression (referred to as null [PSR~0]) (*Figure 2—figure supplement 2*). Gene Ontology analysis of direct SRSF1 translational targets revealed an enrichment in mRNAs associated with cell cycle and chromosome organization, as previously seen when analyzing all targets or CLIP-positive targets (compare *Figure 2D* and *Figure 2—figure supplement 1A,C*) as well as an enrichment in mRNAs linked with transcription and RNA metabolism (*Figure 2D,E*, *Figure 2—figure supplement 1C*).

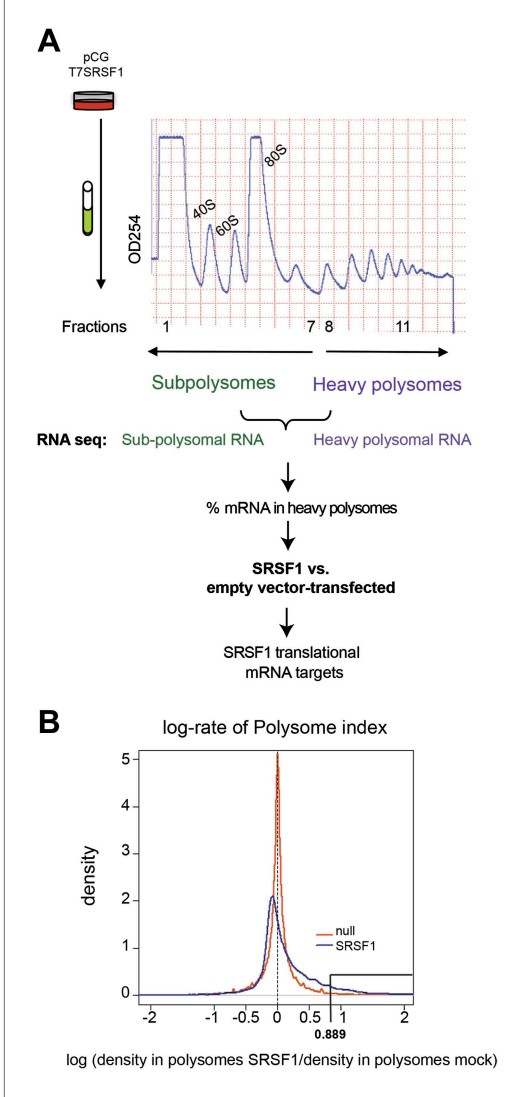

**Figure 1**. Identification of SRSF1 mRNA translational targets. (**A**) Experimental approach to identify SRSF1 mRNA translational targets. A characteristic fractionation profile of empty vector (pCG) and SRSF1 transfected-HEK 293T cells (*Figure 1—figure supplement 1*) is depicted. Absorbance at 254 nm was monitored. (**B**) A plot showing the distribution of mRNAs from RNA-seq analysis according to the polysome shift ratio (PSR). The null distribution (comparing two control subsamples) is symmetric and sharply centered at 0. The PSR of SRSF1 versus empty vector shows an enrichment over the null distribution. The mRNAs with a p<0.01 (PSR>0.889) were considered SRSF1 translational targets (Supplementary file 1).

The following figure supplements are available for figure 1:

**Figure supplement 1**. Optimization of SRSF1 transient transfection for polysomal shift analysis.

**Figure supplement 2**. Luciferase reporter containing an SRSF1 binding site.

## Validation of SRSF1 translational targets

We proceeded to validate a subset of SRSF1 targets that were identified in the experiment described above. These experiments were performed in polysomal/subpolysomal fractions obtained independently from the samples used in the RNA-seq experiment (*Figure 1*). We selected mRNAs encoding for proteins involved in cancer-related pathways such as cell cycle and apoptosis, as well as other targets involved in processes such as transcription, translation, RNA processing, and proteolysis. Moreover, the selected mRNAs covered a wide range of values for PSR, from 0.9186 to 2.95 (data in Supplementary file 1 at Dryad: *Maslon et al. (2014)*). As a negative control we used the AVEN and ALAS1 mRNAs, as their distribution along the polysome profile did not change upon SRSF1 overexpression (PSR = 0) and its cellular abundance was comparable to those mRNAs selected for the validation (data in Supplementary file 1 at Dryad: *Maslon et al. (2014)*). Importantly, we observed a significant increase in the polysomal to subpolysomal ratio for 70% of those mRNAs upon SRSF1 overexpression, as analyzed by RT-qPCR (*Figure 3A*), confirming a role for SRSF1 in regulating translation of these targets. Notably, we observed the highest fold change in the polysome to subpolysome ratio for mRNAs involved in cell cycle regulation and RNA processing.

We sought to further explore a role for SRSF1 in mediating the translational regulation of proteins involved in different aspects of RNA metabolism, such as splicing, NMD, and translation. As a control we included a chimeric SRSF1 protein harboring a nuclear retention signal (NRS), identified in the non-shuttling protein SRSF2, which was fused at its C-terminus. This protein, termed SRSF1-NRS, is constitutively retained in the nucleus and does not activate translation (*Cazalla et al., 2002*; *Sanford et al., 2004*). In most cases, we confirmed that overexpression of SRSF1 protein, but not of the nuclear-retained SRSF1-NRS variant, results in a shift of the mRNAs encoding for the aforementioned proteins to polysomes (*Figure 3B*). For instance, we noticed that overexpression of SRSF1 results in increased translation of CWC22, an essential splicing factor that also has a role in exon junction complex deposition and NMD (*Alexandrov et al., 2012*; *Barbosa et al., 2012*; *Steckelberg et al., 2012*), as well as of PRPF18 that is required for the second step of pre-mRNA splicing (*Horowitz and Krainer, 1997*). SRSF1 translational targets that were

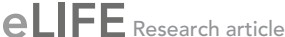

**Figure 2**. SRSF1 translational targets. (**A**) Motif identified on putative direct translational targets using 5-mers enrichment in combination with MEME algorithm (width = 10, sites = 508/508, E value = 1.5E-308, IC = 7.9 bits). Over-represented k-mers were obtained by double comparison CLIP+ versus CLIP− translational targets and CLIP+ translational targets versus null CLIP+. (**B**) Box plot showing the density of consensus motif (translational targets CLIP+ > translational targets CLIP− > null CLIP+ > null CLIP+). (**C**) Venn diagram showing the overlap (505 mRNAs) between translational targets (1576 mRNAs with p<0.01) and CLIP-tag mRNAs containing the consensus motif

*Figure 2. Continued on next page*

*Figure 2. Continued*

(CLIP+ CM) (6065) (Fisher's exact test: OR 1.1686; p<2.2E-16. (**D**) The most representative classes of Gene Ontology terms enriched in direct translational targets (with CLIP-tag and consensus motif [CLIP+ CM]) relative to all the mRNAs detected in HEK 293T by RNA-seq. The number of genes observed in each category is indicated in the pie chart. Modified Fisher's exact p value, EASE score is given for each category. In all cases, the Benjamini–Hochberg-corrected EASE score was <0.1. (**E**) Table giving the gene names of SRSF1 translational targets related to cell cycle and RNA processing pathways.

The following figure supplements are available for figure 2:

**Figure supplement 1**. Analysis of SRSF1 translational targets.

**Figure supplement 2**. The positional bias of the SRSF1 consensus motif in the 5′UTR, protein coding sequence (CDS), and 3′UTR of SRSF1 translational mRNA targets (red) was compared to the null population (blue, PSR~0).

validated in this assay also include LSM3, which is a constituent of the Lsm1-7-Pat1 complex that functions in the 5′-to-3′ mRNA decay pathway (*Sharif and Conti, 2013*), as well as proteins involved in the NMD pathway such as UPF2 and PNRC2 (data in Supplementary file 1 at Dryad: *Maslon et al. (2014)* and *Figure 3B*) (*Nicholson et al., 2010*). In particular, PNRC2 shows a drastic movement to polysomal fractions upon SRSF1 overexpression, but is not responsive to increased expression of the SRSF1-NRS variant (*Figure 3B*). Together, these observations are consistent with a role for SRSF1 in regulating the translation of mRNAs encoding components of the RNA processing pathway. Of interest, we noticed that SRSF1 also promotes the translation of mRNAs encoding negative regulators of mRNA translation, such as EIF4E3 that recognizes and binds the 7-methylguanosine-containing mRNA cap during an early step in translation initiation (*Osborne et al., 2013*), as well as Paip2 that inhibits translation both in vitro and in vivo by displacing PABP from the poly(A) tail (*Khaleghpour et al., 2001*). This could represent a feedback mechanism that becomes activated in response to SRSF1 overexpression to antagonize its role in translation. Alternatively it could suggest a role for SRSF1 in the negative regulation of translation of subsets of mRNAs that are targets of EIF4E3 and/or PAIP2. Indeed, we observed that 165 mRNAs are translationally repressed by increased expression of SRSF1 (*Figure 1B*).

It is known that protein levels in the cell cannot be always predicted from the mRNA abundance, as other factors, such as post-translational modification and protein stability, contribute to steady-state levels of protein. Thus, we sought to determine whether the SRSF1-induced polysomal shift of target mRNAs correlated with higher protein abundance. Use of stable isotope labeling by amino acids in cell culture (SILAC) resulted in the identification of 2157 proteins in the three protein lysates used (untransfected, empty vector, and SRSF1-transfected HEK 293T cells) (data in Supplementary file 2 at Dryad: *Maslon et al. (2014)*). Following normalization, we calculated for each of those proteins the $\log_2$ of the ratio between the levels of each individual protein in cells overexpressing SRSF1 versus control cells (SILAC index). Thus, a positive value of the SILAC index indicates an increase in protein abundance upon SRSF1 overexpression. As expected, we found the higher score for SRSF1 (data in Supplementary file 2 at Dryad: *Maslon et al. (2014)*). To establish a correlation with the PSR, the SILAC index for each protein was assigned to the mRNAs encoded by the corresponding gene (data in Supplementary files 1 and 2 at Dryad: *Maslon et al. (2014)*). We found a positive correlation showing increased protein levels for SRSF1 direct translational targets (*Figure 3C*, third box). This correlation was even better for a subset of mRNAs that displayed a high polysome shift ratio (PSR>1) (*Figure 3C*, fourth box).

## SRSF1 activates translation via the mTOR pathway

We previously showed that SRSF1 promotes translation initiation of bound mRNAs by suppressing the activity of 4E-BP, a competitive inhibitor of cap-dependent translation. This activity is mediated by interactions of SRSF1 with components of the mTOR signaling pathway. This suggested a model whereby SRSF1 functions as an adaptor to recruit signaling molecules responsible for regulation of cap-dependent translation of specific mRNAs (*Michlewski et al., 2008*). We sought to determine whether endogenous SRSF1 translational targets responded to the same mechanism of translational activation, as we previously showed using reporter assays. For this, we treated cells transiently expressing SRSF1 (or control cells) with a specific inhibitor of the mTOR kinase, PP242 (*Dowling et al., 2010*), and then measured

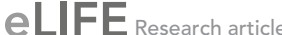

**Figure 3**. Validation of SRSF1 translational targets. (**A**) RT-qPCR validation confirms an increased polysome to subpolysome ratio for selected SRSF1 translational targets upon SRSF1 overexpression (CLIP+: green; CLIP+ and harboring an SRSF1 consensus motif (CLIP+ CM+): purple). The polysome to subpolysome ratio is relative to cells transfected with empty vector (pCG) and normalized to actin. Plotted data are the average of three biological

*Figure 3. Continued on next page*

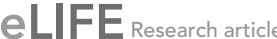 

*Figure 3. Continued*

replicates. The asterisks indicate statistical significance (p<0.05) Error bars, Gene Ontology terms and cancer relationship are indicated. (**B**) RT-qPCR validation of SRSF1 translational targets involved in RNA metabolic processes. The polysome to subpolysome ratio as measured by RT-qPCR in empty vector-transfected cells compared to cells overexpressing SRSF1 is indicated. Two different concentrations of pCGT7-SRSF1 plasmid were used. A mutant version of SRSF1 that is constitutively nuclear and does not activate translation was also included (SRSF1-NRS). (**C**) Box plot showing the values of the stable isotope labeling by amino acids in cell culture (SILAC) index, defined as $\log_2$ SRSF1 ratio/empty vector ratio. 'All' refers to mRNAs encoding for all the proteins found by SILAC (2471 mRNAs); 'CLIP+' refers to the SRSF1 translational targets (PSR>0.889536; p<0.01) harboring CLIP-tag (125 mRNAs); 'CLIP+CM' is for direct translational targets (105 mRNAs); 'PSR≥1, CLIP+CM' refers to direct translational targets with a PSR≥1 (72 mRNAs) (p = 0.002487). (**D**) PP242-mediated mTOR inhibition suppresses SRSF1-dependent activation of translation of a subset of mRNA targets. Control and SRSF1-overexpressing cells were treated with PP242 for 90 min. The polysome to subpolysome ratio was measured by RT-qPCR in empty vector-transfected cells compared to cells overexpressing SRSF1 treated with or without PP242. mTOR inhibition was validated by Western blotting (data not shown).

polysomal to subpolysomal ratios of a subset of selected SRSF1 translational targets, including proteins related to RNA processing (*Figure 3B*). Interestingly, we found that inhibition of mTOR abrogated the stimulatory activity of SRSF1 on the translation of selected targets (*Figure 3D*). This demonstrates that the activity of SRSF1 in translational activation of endogenous targets requires the mTOR pathway.

## Coupling of alternative splicing and translation

The function of shuttling SR proteins in both splicing and post-splicing activities raises the possibility that they may act to coordinate nuclear and cytoplasmic events for a subset of pre-mRNAs. Indeed, we previously showed by coupling CLIP with subcellular fractionation that mRNAs found associated with SRSF1 in the nucleus, were also found in the cytoplasm and in the actively translating pool of ribosomes, suggesting that splicing and translation of those mRNAs could be coordinated by SRSF1 (*Sanford et al., 2008*). To assess a global effect of SRSF1 in coupling of pre-mRNA splicing with mRNA translation, we analyzed changes in alternative splicing in cells overexpressing SRSF1 using exon-junction arrays. We identified 382 differentially regulated cassette exons: 209 events associated with skipping of an alternative exon and 173 events where overexpression of SRSF1 resulted in the inclusion of an alternative exon (data in Supplementary file 3 at Dryad: *Maslon et al. (2014)*). In order to correlate the polysomal shift of SRSF1 translational targets with the alternative splicing of those mRNAs, the PSR for the isoforms generated by the changes in alternative splicing events upon SRSF1 overexpression were analyzed (*Figure 4A*). Interestingly, we observed a statistically significant increase in the PSR of the isoforms generated by skipping as well as inclusion of a cassette exon (*Figure 4A*). This suggests that SRSF1 can influence both the alternative splicing as well as the translational efficiency of subsets of mRNAs. As an example, we focused on the alternative splicing of the SR protein kinase Clk1 pre-mRNA, which is regulated through alternative splicing, giving rise to two isoforms encoding catalytically active and truncated inactive polypeptides (Clk1Ex4+ and Clk1Ex4−, respectively) (*Duncan et al., 1997*). Indeed, we could confirm that SRSF1 overexpression caused increased inclusion of the alternatively spliced exon 4 of CLK1 mRNA, giving rise to the active Clk1 isoform (*Figure 4B*, left panel). Furthermore, an SRSF1 CLIP tag containing a consensus motif mapped to this cassette exon suggesting that it is bound directly by SRSF1 (data in Supplementary file 3 at Dryad: *Maslon et al. (2014)*). Interestingly, this isoform was also more translated (as measured by polysome to subpolysome ratio by RT-qPCR) upon SRSF1 overexpression (*Figure 4B*, right panel).

## A role for SRSF1 translational activity in cell division

A large number of SRSF1 mRNA translational targets encode for proteins involved in cell cycle regulation (data in Supplementary file 1 at Dryad: *Maslon et al. (2014)*). This is even more apparent when looking at the functional classification of direct translation targets containing the SRSF1 consensus binding motif (*Figure 2D,E*) that includes several redundant layers such as cell cycle (p = 3.35E-06), chromosome organization (p = 2.79E-07), and M phase (p = 3.61E-05), suggesting that SRSF1-mediated translation may affect proper mitotic progression. In particular, many of these mRNAs encode for proteins with a role in mitotic spindle and kinetochore formation (data in Supplementary file 1 at Dryad: *Maslon et al. (2014)* and *Figure 5A*). The aforementioned group of proteins comprises, among others, a group of centrosomal proteins including CEP170, CEP70, and CEP57, proteins involved in kinetochore and spindle function, such as NDC80 and CCDC99, and proteins forming the condensin complex, including SMC2 and SMC4. NDC80, a core protein of the NDC80 complex, is

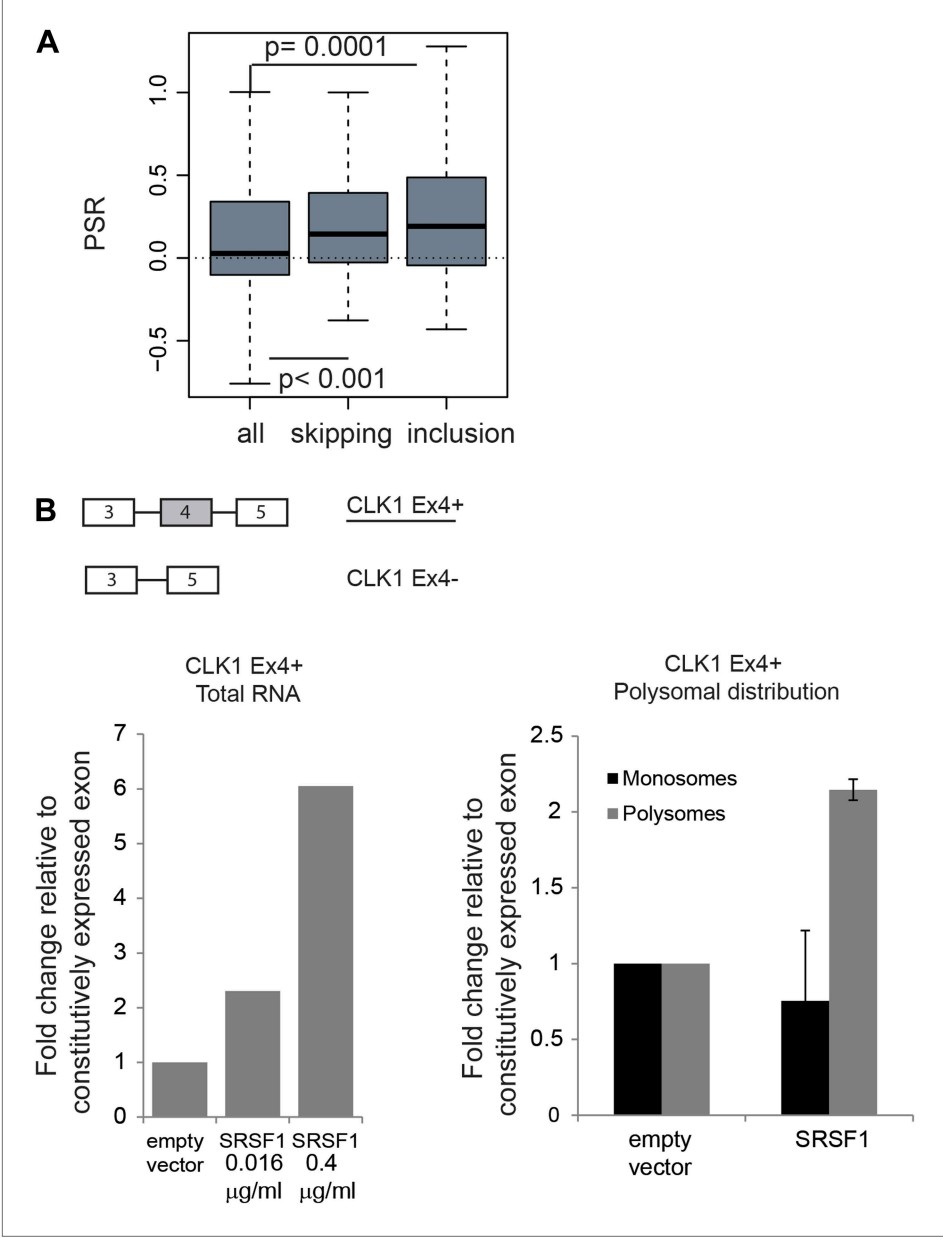

**Figure 4**. Coupling of alternative splicing and translational regulation. (**A**) Correlation between SRSF1-induced changes in alternative splicing with polysomal distribution of those isoforms. Changes in alternative splicing induced by SRSF1 overexpression were determined by an exon-junction array. PSR: polysome shift ratio. (**B**) RT-qPCR analysis of the effect of SRSF1 on CLK1 alternative splicing and preferential polysomal association. The exon-intron structure of both isoforms is indicated (not to scale) and the CLK1 isoform that is an SRSF1 direct translational target is underlined. SRSF1-induced changes in CLK1 alternative splicing were determined and normalized to exon 7 (constitutive exon) levels (left panel). Two different concentrations of pCGT7-SRSF1 plasmid were used. Polysomal distribution of CLK1 mRNA isoforms upon SRSF1 overexpression normalized to exon 7 (constitutive exon) levels (right panel).

required for stable microtubule binding in the outer plate of kinetochores (**Wei et al., 2007**), whereas CCDC99, also known as Spindly, is required for the dynein/dynactin localization to kinetochores (**Barisic et al., 2010**). Centrosomal proteins control cell cycle progression and spindle–kinetochore assembly (**Kumar et al., 2013**). In particular, CEP57 is involved in linking central spindle microtubules and is required for spindle integrity (**He et al., 2013**), CEP70 is necessary for the organization and

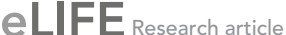

**Figure 5**. SRSF1 translational targets involved in cell division. (**A**) List of cell cycle proteins regulated by SRSF1 at the translational level (left panel). The cartoon depicting their involvement in chromosome segregation during mitosis was adapted from **Kitagawa and Hieter (2001)**. APC stands for (Anaphase-promoting complex) (**B**) Validation of cell cycle translational targets. The polysome to subpolysome ratio for a subset of cell cycle-related mRNAs was measured by RT-qPCR in empty vector-transfected cells compared to cells overexpressing SRSF1 or depleted of SRSF1. Two different concentrations of pCGT7-SRSF1 plasmid were used. A mutant version of SRSF1 that is constitutively nuclear and does not activate translation was also included (SRSF1-NRS). The asterisks indicate statistical significance (p<0.05). (**C**) Western
*Figure 5. Continued on next page*

*Figure 5. Continued*

blot validation of selected cell cycle SRSF1 translational targets in empty vector or SRSF1-transfected cells. β-Actin was used as a loading control. (**D**) Western blot validation of selected cell cycle SRSF1 translational targets in control or SRSF1-depleted cells in an asynchronous (**A**) or mitotic population (**M**). Tubulin was used as a loading control.
The following figure supplements are available for figure 5:

**Figure supplement 1**.

orientation of a bipolar spindle in mitosis, and CEP170 is involved in microtubule organization and associates with spindle microtubules during mitosis (**Shi et al., 2011**). Finally, SMC2 and SMC4 proteins are part of the condensin I and II complexes, which, together with cohesin, restructure chromosomes to promote faithful chromosome segregation during mitosis (**Losada and Hirano, 2005**). Importantly, altered translational regulation of any of these proteins could have important implications for faithful chromosome segregation, as this depends on the formation of a bipolar spindle and the correct attachment of kinetochores to spindle microtubules. Notably, another identified SRSF1 translational target, CDK1, has also been shown to regulate the assembly of mitotic spindles as well as spindle positioning, stability, and elongation (**Enserink and Kolodner, 2010**).

Here, we focused on those SRSF1 translational targets that are related to chromosome segregation during mitosis (**Figure 5A**). We estimated their polysome to subpolysome ratio in HEK 293T cells over-expressing SRSF1 using RT-qPCR. We observed a shift to the polysomal fraction upon increased SRSF1 expression (note that both concentrations shown in **Figure 1—figure supplement 1** were used here). By contrast, increased expression of the translationally inactive SRSF1-NRS variant did not cause a shift to the polysomal fraction in most cases. Also, as expected we observed a decrease in polysomal association of these mRNAs upon siRNA-mediated depletion of SRSF1 (**Figure 5B**). This analysis validated this subset of mRNAs as bona fide translational targets, suggesting a role for SRSF1 in the translational regulation of proteins that are required for cell cycle progression (**Figure 5B**). We also performed Western blotting analysis of some of these targets and were able to confirm that SRSF1 overexpression results in increased levels of CEP70, NDC80, and SMC4 (**Figure 5C**). Conversely, we observed decreased protein levels following SRSF1 depletion, in particular of NDC80, SMC4, and CEP57 (**Figure 5D**). Furthermore, SILAC analysis also showed that increased expression of SRSF1 resulted in increased levels of proteins related to cell cycle progression and chromosome segregation (**Figure 3** and data in Supplementary file 2 at Dryad: **Maslon et al. (2014)**). In particular, CEP170, CBX3, and PDS5B protein abundance increased in response to SRSF1 overexpression, further validating the role of SRSF1 in regulating translation of these targets. In agreement with these observations, previous findings from the MitoCheck consortium have revealed that SRSF1 is involved in mitotic progression (**Neumann et al., 2010**).

A recent study has revealed a dynamic reprogramming of translation throughout the cell cycle (**Stumpf et al., 2013**). Since our studies were carried out for the most part in asynchronous cell populations, it remains possible that SRSF1 non-translational effects on cell cycle could be the cause of at least some of the observed changes in mRNA translation. Importantly, we observed that increased expression of SRSF1 does not grossly affect the cell cycle profile (**Figure 5—figure supplement 1A**), strongly suggesting that the observed increase in polysomal to subpolysomal ratios for these targets is primarily linked to SRSF1-mediated translation and not to an indirect effect on the cell cycle. Conversely, SRSF1 depletion led to major cell cycle aberrations (**Figure 5—figure supplement 1B**), with approximately 50% of SRSF1-depleted cells remaining arrested in the G2/M phase. This could be caused by a loss of SRSF1-dependent translation of targets required for cell cycle progression, as we observed decreased association of these mRNA targets with polysomes in SRSF1-depleted cells (**Figure 5B**), as well as decreased levels of the corresponding proteins (**Figure 5D**). Importantly, the levels of SRSF1 itself do not change significantly throughout the cell cycle, albeit there is a 1.2-fold increase in SRSF1 protein levels from the G1 to S phase (**Ly et al., 2014**). We cannot rule out, however, that the subcellular localization of SRSF1 is cell cycle regulated, which could potentially affect the translation of SRSF1 targets.

We proceeded to further assess the effect of SRSF1 depletion in mitotic progression. The proteasome inhibitor MG132 was added for 2 hr to HeLa cells in culture and the cell cycle profile was evaluated at different times following MG132 release (**Figure 6A**). As previously demonstrated, proteasome

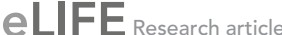

**Figure 6**. SRSF1 is required for cell cycle progression. (**A**) and (**C**) Schematic representation of the protocols used to assess mitotic progression. (**B**) HeLa cells were treated as in (**A**) and the number of cells at different stages of mitosis was determined by classification of images of fixed cells stained for DNA, tubulin, and pericentrin. (**D**) HeLa cells were treated as in (**C**) and time-lapse imaging of mCherry-H2B was performed. Images were captured every 15 min over 24 hr at three different positions. Representative images of cells transfected with control siRNA and SRSF1 siRNA are shown. (**E**) The graph indicates the elapsed time (minutes) from nuclear envelope breakdown (NEBD) or chromatin condensation to the onset of anaphase/telophase or to mitotic cell death. (**F**) HT1080 cells stably expressing GFP-CENPA were transfected with control or SRSF1-specific siRNA, and 24 hr later cells were retransfected with either empty vector or SRSF1. The next day cells were seeded in six-well plates and time-lapse imaging of GFP-CENPA was performed as in (**C**). The graph indicates the elapsed time (minutes) from nuclear envelope breakdown (NEBD) or chromatin condensation to the onset of anaphase/telophase or to mitotic cell death. Scale bar is 10 µm. The asterisks indicate statistical significance (*p<0.05, **p<0.01, ***p<0.001).

inhibitors induce metaphase arrest (*Wójcik et al., 1996*) and indeed following drug treatment we observed accumulation of cells in the prometaphase/metaphase stage of the cell cycle (*Figure 6B*). Interestingly, while control cells proceeded through the normal stages of mitosis following MG132 withdrawal, HeLa cells that were depleted of SRSF1 remained arrested in metaphase, indicating that SRSF1 was required for normal mitotic progression (*Figure 6B*). To confirm this, we co-transfected HeLa cells with epitope-tagged GFP-tubulin and mCherry-H2B and compared the cell cycle stage of control cells and SRSF1-depleted cells by time-lapse imaging (*Figure 6C–E*). The co-transfected proteins were used as markers to follow the cell cycle stage (for simplicity only the mCherry-H2B is shown). This analysis confirmed that SRSF1 is indeed essential for proper cell cycle progression (*Figure 6D,E*). Specifically, control cells underwent mitosis in around 50 min, whereas cells depleted of SRSF1 remained arrested in metaphase for several hours, and eventually either underwent cell division or apoptosis. We repeated this experiment in HT1080, a human fibrosarcoma cell line stably expressing GFP-centromere protein A (CENPA). CENPA is homologous to histone H3 and replaces canonical H3 in the nucleosome core of centromeric chromatin. Thus, monitoring GFP-CENPA protein allows progression through mitosis to be followed. Similarly to what was observed in HeLa cells, SRSF1 depletion in HT1080 resulted in a significant, albeit less severe, increase in the time these cells spent in mitosis. Interestingly, the mitotic delay was partially rescued by restoring normal levels of SRSF1 (*Figure 6F*).

The finding that SRSF1 translational targets are enriched for mRNAs implicated in mitotic spindle function could explain the observed mitotic defect. To explore this further, we followed spindle formation in control cells and in cells depleted of SRSF1. We noticed that upon depletion of SRSF1, HeLa cells displayed spindle defects, in particular a multipolar spindle phenotype (*Figure 7A–C*). SRSF1 depletion also resulted in abnormal alignment and chromosome congression problems (*Figure 7—figure supplement 1*). Importantly, the multipolar spindle phenotype could be rescued by transient overexpression of wild-type SRSF1 protein. By contrast, transient expression of the non-shuttling SRSF1-NRS did not rescue this phenotype (*Figure 7D*). This strongly implies that normal levels of SRSF1 protein are required to maintain a bipolar spindle and that the translational function of SRSF1 is necessary for this activity.

In summary, we have identified the translational targets of the shuttling SR protein SRSF1 and have found a particular enrichment in mRNAs that participate in cell cycle regulation and chromosome segregation. In particular, we identified a role for SRSF1 in translating mRNAs encoding for proteins involved in bipolar spindle formation. The translational regulation of SRSF1 targets could contribute partially to the role of this shuttling SR protein in tumorigenesis.

## Discussion

There is extensive coupling among different steps in eukaryotic gene expression, as illustrated by the intimate connection between transcription and pre-mRNA splicing (*Baurén and Wieslander, 1994*; *Neugebauer, 2002*). Recent genome-wide analyses using the CLIP protocol have identified endogenous RNA targets for several SR proteins (*Sanford et al., 2009*; *Änkö et al., 2012*; *Pandit et al., 2013*). These studies revealed that individual mRNAs bind multiple SR proteins, as was also described in insect cells (*Björk et al., 2009*).

Protein synthesis is a tightly regulated process and its misregulation has been linked to the development of cancer (*Blagden and Willis, 2011*). We had previously demonstrated that SRSF1 functions as an adaptor protein to recruit the signaling molecules responsible for the regulation of cap-dependent translation of specific mRNAs (*Michlewski et al., 2008*). Here, we present a transcriptome-wide view of the role of SRSF1 in mRNA translation. This analysis revealed that SRSF1 regulates the translation of mRNAs encoding proteins involved in many different cellular processes, including cell cycle progression, RNA processing, and mRNA translation itself (*Figure 3A*). For instance, SRSF1 promotes the translation of PABP-interacting protein 1 (PAIP1), a positive regulator of translation that binds to eIF3 and stabilizes the interaction between PABP and eIF4G (*Martineau et al., 2008*). The activity of SRSF1 in promoting the translation of endogenous targets is mTOR dependent; however, we did not find an effect of SRSF1 on the regulation of either 5′TOP mRNAs or pyrimidine rich translational element (PRTE) mRNAs, which have been previously shown to be mTOR sensitive (*Hsieh et al., 2012*; *Thoreen et al., 2012*). Interestingly, approximately one third of the SRSF1 translational target mRNAs identified here were previously shown to be bona fide RNA targets of this SR protein by CLIP-seq, suggesting that these are direct translational targets (*Sanford et al., 2009*). Recently, an interesting link between alternative splicing and the preferential association of alternative mRNA isoforms to the translational

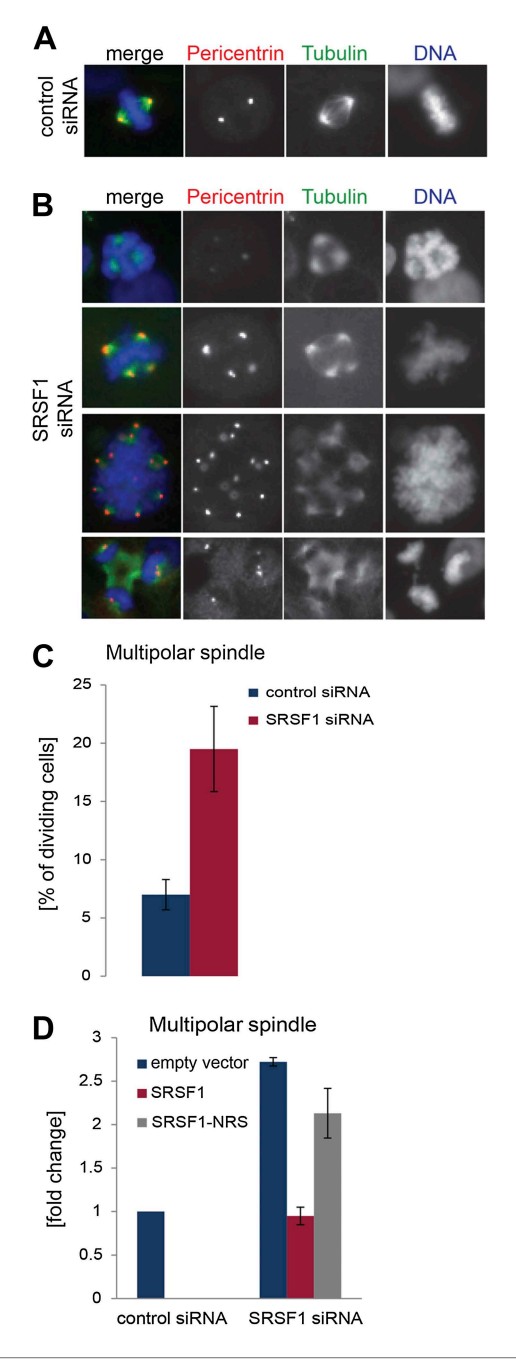

**Figure 7**. SRSF1 is required for bipolar spindle formation. (**A** and **B**) HeLa cells were transfected with control or SRSF1-specific siRNA and 48 hr later stained for DNA (blue), tubulin (green), and pericentrin (red). Representative images for cells transfected with control or SRSF1-siRNAs (**A** and **B**, respectively) are shown. (**B**) Images show the formation of multipolar spindles upon SRSF1 depletion. (**C**) Quantitation of the multipolar spindle phenotype observed in (B) upon SRSF1 depletion. (**D**) HeLa cells were treated as in (**A** and **B**) and 24 hr later transfected with either wild-type SRSF1 or its

*Figure 7. Continued on next page*

machinery was reported (*Sterne-Weiler et al., 2013*). Here, we have analyzed changes in alternative splicing in response to different levels of SRSF1 protein. Our previous results have suggested that SRSF1 may act to coordinate the nuclear and cytoplasmic steps of post-transcriptional gene expression for a subset of pre-mRNAs (*Sanford et al., 2008*). Here, we observed that mRNAs that display alternative splicing changes upon increased SRSF1 concentration were also translationally regulated by SRSF1. This suggests that SRSF1 influences several steps of the mRNA life cycle acting to couple nuclear events with mRNA translation (*Figure 4*). Interestingly, it was recently shown that SRSF3 regulates the alternative splicing of programmed cell death 4 (PDCD4) mRNA, but also negatively regulates its translational efficiency in the cytoplasm (*Kim et al., 2014*).

SRSF1 was previously reported to be required for the maintenance of genomic stability in chicken DT40 cells and its inactivation results in a G2-phase cell cycle arrest and subsequent programmed cell death (*Li and Manley, 2005*; *Li et al., 2005*).

In this study, we show that SRSF1 is required for normal mitotic progression and its depletion results in metaphase arrest (*Figure 6*) and the formation of a multipolar spindle, which in general is not compatible with cell survival (*Figure 7*). Along the same lines, loss of another SR protein, SRSF2 (SC35) in mouse embryonic fibroblasts also results in a G2/M cell cycle arrest and genomic instability (*Xiao et al., 2007*). SRSF1 and SRSF3 associate with interphase chromosomes and post-mitotic chromatin; however, during mitosis they are displaced from chromatin by phosphorylation of histone H3 on serine 10 (*Loomis et al., 2009*). Interestingly, SRSF1 mRNA is deposited onto mitotic microtubules; however, the regulation of its translation during cell cycle is not currently understood (*Blower et al., 2007*). Moreover, SRSF1 has been recently identified as a factor involved in centriole biogenesis, another mechanism essential for mitosis and genomic integrity. These findings suggest a potential role for SRSF1 in regulating chromatin function and cell cycle progression (*Balestra et al., 2013*).

The oncogenic potential of SR proteins has so far being described in terms of the nuclear activities of SR proteins, most notably involving differential alternative splicing of pre-mRNAs involved in signaling and/or cellular transformation (*Ghigna et al., 2005*; *Karni et al., 2007*). A recent study confirmed that the transformative potential of SRSF1 requires its splicing activity,

*Figure 7. Continued*

nuclear-retained version (SRSF1-NRS). The next day cells were fixed and stained for DNA, tubulin, and pericentrin and the appearance of multipolar spindle was quantified. Scale bar is 5 µm.
The following figure supplements are available for figure 7:

**Figure supplement 1**. SRSF1 is required for appropriate chromosome alignment.

since deletion of its first RRM (RRM1), which is required for pre-mRNA splicing, abrogates this activity. Nonetheless, this study also revealed that preventing the nucleo-cytoplasmic shuttling of SRSF1 prevents its oncogenic potential, strongly suggesting a role for the translational activity of SRSF1 in its oncogenic activities (*Shimoni-Sebag et al., 2013*). Interestingly, SRSF1 as well as SRSF9 (SRp30c) have been recently shown to promote the translation of β-catenin in an mTOR-dependent manner and this activates Wnt signaling-mediated tumorigenesis. In agreement, SRSF9 displays oncogenic properties and its overexpression has been observed in multiple types of human tumors (*Fu et al., 2013*). Results presented here strongly suggest that overexpressed SRSF1 could contribute to tumorigenesis by influencing the translational rate of key components of the cell cycle machinery. Indeed, many of the SRSF1 translational targets that we have identified here correspond to proteins with roles in chromosome segregation and cell cycle progression (data in Supplementary file 1 at Dryad: *Maslon et al. (2014)*, *Figure 2* and *Figure 5*). In particular, we found that SRSF1 has a role in regulating the translation of proteins that are required for mitotic spindle and kinetochore function. Importantly, altered expression of any of the spindle-associated proteins may contribute to unbalanced chromosome segregation during mitosis. Indeed, increased expression of NDC80 and SMC2 has been observed in cancer cells and an increased requirement for NDC80 at kinetochores of cancer cells has been postulated (*Ferretti et al., 2010*; *Dávalos et al., 2012*). Interestingly, some SRSF1 translational targets, including NDC80 and centrosomal proteins, have been previously identified in an siRNA screen looking for proteins involved in centrosome clustering in cancer cells (*Leber et al., 2010*). This suggests that the oncogenic function of SRSF1 could be partially due to its role in promoting the translation of factors that are involved in suppressing multipolar divisions in human tumor cells. Indeed, decreased levels of SRSF1 protein result in multipolar spindle formation and abnormal chromosomal alignment (*Figure 7—figure supplement 1*). Importantly, we were able to rescue the multipolar spindle phenotype observed upon SRSF1 depletion only with overexpression of the wild-type SRSF1 protein. By contrast, increased expression of the nuclear-retained SRSF1-NRS did not rescue this phenotype, strongly suggesting a role for SRSF1-regulated translation in this process (*Figure 7*).

So far, most studies on gene regulation during cell cycle progression have focused on the transcriptional regulation of mRNAs encoding proteins required for this process as well as on the timely proteasome-mediated degradation of checkpoint proteins. Lately however, a crucial role for the translational regulation of hundreds of mRNAs required for cell cycle progression, including most of the mRNAs encoding proteins forming the cohesin and condensin complexes, has been uncovered (*Stumpf et al., 2013*). Interestingly, and despite the fact that our experiments were performed in an unsynchronized cell population, we observed an approximate 15% overlap between SRSF1 direct translational targets identified in this work and those genes that were shown to exhibit translation regulation during cell cycle progression (*Stumpf et al., 2013* and this study), strongly suggesting that SRSF1 may have a central role in this event (Fisher's exact test: OR 1.466548, p=0.02611). Those overlapping targets include the condensin components, SMC2 and SMC4, the centrosomal protein CEP170, and the DNA repair protein MRE11A. Altogether, this suggests that SRSF1 could provide a transcript-specific mechanism for translational regulation of the cell cycle. Increased expression of SRSF1 would promote the increased translation of genes associated with cell division and this could partially explain the oncogenic role of SRSF1. In summary, these data provide insights on the complex role of SRSF1 in the control of gene expression and its implications in cancer.

## Materials and methods

### Cell culture and reagents

HEK 293T, HeLa, and HT1080 cell lines were grown in Dulbecco's Modified Eagle's Medium (Invitrogen) supplemented with 10% fetal calf serum, and incubated at 37°C in the presence of 5% $CO_2$. Control pooled siRNA (D-001810-01), SRSF1 pooled siRNA (L-018672-01), and SRSF1 UTR-targeting siRNA

(J-018672-12) were purchased from Thermo Scientific. Cycloheximide was from Merck Chemicals and used at 50 µg/ml. PP242 was from Cayman Chemicals and was used at 5 µM.

## Cell synchronizations
Cells were grown to 30% confluency and then incubated with 2 mM thymidine (VWR International) for 18 hr, washed with PBS, and released into thymidine-free medium for 9 hr. Thymidine (2 mM) was then added for a further 16 hr. The cells were then washed with PBS, released into thymidine-free medium, and harvested at different time points, as indicated in the figure legend (*Figure 5—figure supplement 1*), and analyzed by propidium iodide-flow cytometric analysis.

## Cell cycle analyses
Following the indicated treatment, cells were collected by centrifugation (1000 rpm, 4 min). After washing with PBS solution, cells were fixed with chilled 70% ethanol at 4°C for 24 hr. The cells were then centrifuged (1000 rpm, 4 min), washed once with PBS solution, resuspended in PBS, incubated with 5 µl of RNase A (0.5 µg/ml; Roche) for 30 min at 37°C, and stained with 50 µg/ml propidium iodide (Sigma) for 30 min at room temperature. Cell cycle distribution was then evaluated using flow cytometry.

## Epitope-tagged expression plasmids
The mammalian expression vector pCGT7-SRSF1 (previously known as pCG T7-SF2/ASF) has been previously described (*Cáceres et al., 1997*). Transcription is driven by the cytomegalovirus enhancer-promoter and the coding sequence begins with an N-terminal epitope tag, MASMTGGQQMG; this sequence corresponds to the first 11 residues of the bacteriophage T7 gene 10 capsid protein and is recognized by the T7.tag monoclonal antibody (Novagen). The mammalian expression vector GFP-α-tubulin and RFP-H2B were provided by Carol-Anne Martin (MRC HGU).

## DNA and siRNA transfection
Using Lipofectamine 2000 (Invitrogen), 70–90% confluent cells were transfected with the indicated amount of pCG T7 expression vector. The transfection medium was replaced with fresh medium with 10% FCS after 5 hr and following 24 or 48 hr incubation, cells were harvested and lysed or seeded for subsequent analysis. To determine transfection efficiency, HEK 293T cells were transfected in the same conditions with a plasmid encoding green fluorescent protein (GFP). The efficiency of transfection was measured 72 hr later using a FACS cantoII (BD) flow cytometer. Using DharmaFECT1 reagent (Thermo Scientific) according to manufacturer's protocol, 30–50% confluent cells were transfected with 100 nM siRNA.

## RNA isolation and RT-qPCR
RNA was isolated using TRIzol LS Reagent (Invitrogen) following the manufacturer's protocol. RNA was then treated with Dnase (Ambion) and transcribed to cDNA using the First-Strand Synthesis System from Roche. This was followed by a probe detection qPCR assay (RealTime ready Custom Panel; Roche). For splicing validation and luciferase reporter mRNA analysis, the SYBR Green detection system was used (Lightcycler 2× SYBR Green Mix; Roche). The polysomal to monosomal ratio was calculated using the ΔΔCt method and the statistical analyses were performed using the Mann–Whitney U test.

## Protein extraction, antibodies, and Western blotting
Cell pellets were lysed in 50 mM Tris pH 8.0, 150 mM NaCl, and 1% NP-40 buffer containing protease inhibitors. Protein samples either from HEK 293T or HeLa cell extracts were separated by SDS–PAGE and electroblotted onto nitrocellulose membranes (Whatman) in 25 mM Tris-base, 40 mM glycine, and 20% methanol in a Genie Blotter unit (Idea Scientific Company), at 12 V for 1 hr or iBlot System (Invitrogen) for 6 min. Non-specific binding sites were blocked by incubation of the membrane with 5% non-fat milk in PBS containing 0.1% Tween 20 (PBST). Proteins were detected using the following primary antibodies diluted in blocking solution: mouse monoclonal anti-SRSF1 (clone 96, 1:1000; *Hanamura et al., 1998*), rabbit polyclonal anti-GAPDH (1:2000; Abcam), mouse monoclonal anti-T7 (1:10,000; Novagen), rabbit polyclonal anti-CEP170 (1:1000, Abcam), rabbit polyclonal anti-SMC4 (1:1000; Bethyl Laboratories), rabbit polyclonal anti-CEP70 (1:1000; Abcam), rabbit polyclonal anti-CEP57 (1:250; Abcam), mouse monoclonal anti-NDC80 (1:1000; Abcam), and mouse anti-β-actin (1:5000; Sigma-Aldrich). Following washing in PBST, blots were incubated with the appropriate secondary antibodies conjugated to horse-radish

peroxidase (Pierce) and detected with Super Signal West Pico detection reagent (Pierce). The membranes were stripped using ReBlot Plus Strong Antibody Stripping solution (Chemicon) equilibrated in water, blocked in 5% milk in PBST, and reprobed, as described above.

## Luciferase assay

Cells were transfected with the indicated constructs, including various pGL3 constructs using Lipofectamine 2000. Cells were then lysed on the plate using passive lysis buffer (Promega) and used for the Dual Luciferase Assay Kit following the manufacturer's guidelines (Promega). Samples were measured on a Monolight 3010 luminometer (Pharmingen). Firefly luciferase activity was normalized to Renilla luciferase expression.

## Cell fractionation and sucrose gradient centrifugation

HeLa and/or HEK 293T cell were treated with 50 µg/ml cycloheximide for 30 min at 48 hr after transfection. Cells were subsequently washed twice in ice-cold PBS containing cycloheximide. Cytoplasmic extracts were prepared as previously described (*Sanford et al., 2004*). Sucrose gradients (10–45%) containing 20 mM Tris, pH 7.5, 5 mM MgCl$_2$, and 100 mM KCl were made using the BioComp gradient master. Extracts were loaded onto the gradient and centrifuged for 2.5 hr at 41,000 rpm in a Sorvall centrifuge with a SW41Ti rotor. Following centrifugation, gradients were fractionated using a BioComp gradient station model 153 (BioComp Instruments, New Brunswick, Canada) measuring cytosolic RNA at 254 nm. Fractions 8 to 11 (polysomal fractions) and 1 to 7 (subpolysomal fractions) were pooled and sucrose concentration was adjusted to 20% w/v. The RNA extraction was performed as described above.

## Analysis of RNA-seq reads

All sequence reads were mapped to the RefSeq transcripts (*Pruitt et al., 2009*) using GEM (*Marco-Sola et al., 2012*), allowing for up to three mismatches per read and testing for both strands. Unambiguous reads, mapping to unique positions in the reference, and ambiguous reads, mapping to up to 10 multiple positions, were collected (*Table 1*).

Only the reads mapping forward in transcripts were kept for the analysis (*Table 2*).

For each mRNA (*a*), the density of read counts was calculating using the reads per kilobase per million of mapped reads (RPKM) in a given sample (*N*):

$$d(a, N) = 10^9 \frac{n(a, N)}{N \times length(a)} \tag{1}$$

Using these densities, for each transcript and for each of the two samples, polysomal (*poly*) and subpolysomal (*sub*), a polysomal index (*P*) was defined:

$$P(a) = \frac{d(a, N_{poly})}{d(a, N_{poly}) + d(a, N_{sub})} \tag{2}$$

This index measures the proportion of transcript copies that is present in polysomes.

Then, in order to determine the mRNAs that shift to polysomes upon *SRSF1* overexpression, we defined a polysome shift ratio (*PSR*), as the log$_2$ ratio of the polysomal index between the *SRSF1* overexpressed and mock experiments:

$$PSR = \log_2\left(\frac{P(a, SRSF1)}{P(a, mock)}\right) \tag{3}$$

**Table 1.** Number of sequenced and mapped reads from each sample

| Sample | Total reads | Mapped reads | Unambiguous | Ambiguous |
|---|---|---|---|---|
| Polysomal (mock) | 15,303,461 | 13,225,802 | 8,629,499 | 4,596,303 |
| Polysomal (SRSF1) | 19,135,008 | 15,505,493 | 10,150,775 | 5,354,718 |
| Subpolysomal (mock) | 15,553,058 | 11,676,769 | 7,489,329 | 4,187,440 |
| Subpolysomal (SRSF1) | 14,417,240 | 11,025,797 | 7,106,934 | 3,918,863 |



**Table 2.** Total number of forward read counts considered in each sample

| | Total forward counts |
|---|---|
| Polysomal (mock) | 10,407,102 |
| Polysomal (SRSF1) | 12,219,337 |
| Subpolysosomal (mock) | 9,188,985 |
| Subpolyosomal (SRSF1) | 8,585,825 |

To estimate the cases that change significantly, two non-overlapping subpopulations of read counts from the mock sample were compared to each other. From these two subpopulations, the polysomal index and the ratio between them were calculated. From this comparison, we calculated an empirical p value.

## Comparison with the CLIP-seq data

The SRSF1 mRNA translational targets were compared with RNAs having tags from the CLIP experiments from *Sanford et al. (2008)* and (*2009*). A total of 23,633 CLIP tags obtained from a 454 sequencing experiment in *Sanford et al. (2009)* were mapped to the RefSeq mRNA set using Exonerate (*Slater and Birney, 2005*) with an ungapped alignment model. Sequence tags that fully aligned to the mRNA in the forward strand were kept. A total of 9,094 different mRNAs from RefSeq were found to contain one or more tags.

## SRSF1 translational targets and motif analysis

For the k-mer enrichment analysis, we considered the total count of 5-mers in CLIP-tag regions. The significant differences in relative abundance of 5-mers between the two sets were estimated using the z score statistic (*Fairbrother et al., 2002*):

$$z = \frac{\dfrac{X_A}{N_A} - \dfrac{X_B}{N_B}}{\sqrt{\left(\dfrac{1}{N_A} - \dfrac{1}{N_B}\right)g(1-g)}} \tag{4}$$

where $X_A$ and $X_B$ are the number of occurrences of a given 5-mer in sets A and B, respectively; $N_A$ and $N_B$ are the total number of occurrences of all 5-mers in sets A and B, respectively, and $g=(X_A + X_B)/N_A + N_B)$.

SRSF1 translational targets were defined as those mRNAs that shifted to polysomes significantly, that is PSR>0.889 (TTR, 1576 mRNAs). The set of mRNAs that did not shift were defined as those mRNAs with PSR −0.02<p<0.02 (null, 1448 mRNAs). If two or more mRNAs from a set had 80% or greater sequence similarity, only the longest one was kept. Accordingly, we were left with 1052 and 1133 sequences, respectively. The 24 most significant 5-mers were selected (z score≥35, upper tail ~2.3%), including: AAAAG, AAAAT, AAAGA, AAATA, AAATG, AAATT, AATAT, AATTA, AATTT, AGAAA, ATAAA, ATATT, ATTTA, ATTTT, GAAAA, TAAAA, TAAAT, TATTT, TGAAA, TTAAA, TTATT, TTTAA, TTTAT, and TTTTA.

In order to select candidates for direct targets of SRSF1, we considered the tags from a previous CLIP-Seq experiment (*Sanford et al., 2008*, *2009*). First, all mRNAs were separated into those with CLIP-tags (CLIP+) and those without CLIP-tags (CLIP-). Subsequently, a double comparison of 5-mers was considered:

1. 5-mers inside CLIP-tags in TTR+ CLIP+ mRNAs versus 5-mers inside CLIP-tags in *null CLIP+* mRNAs.
2. 5-mers inside CLIP-tags in TTR+ CLIP+ mRNAs versus 5-mers in TTR+ CLIP+ mRNAs outside the CLIP-tags, that is the rest of the direct targets.

The double ranking of z scores was used to select 5-mers associated with direct translational targets. Subsequently, considering positives z scores such that p≤10⁻⁵ in both rankings, we found 12 GAA-rich 5-mers associated with direct targets: AAAAG, AAAGA, AAAGG, AAGAA, AAGAT, AGAAA, ATGAA, ATTGG, GAAAA, GAAGA, TGGAA, and TTGGA.

In order to infer a consensus motif logo for direct translational targets, we first mapped the selected 5-mers on the mRNA sequences extending 10 nt per flank. The resulting continuous sequences in TTR+ CLIP+ mRNAs were extracted.

The background model for MEME was built using a Markov model with the 'null' sequences for translational targets (M1) and with the CLIP-tags in *null* mRNAs and in TTR+ CLIP+ mRNAs outside the CLIP-tags for the direct translational targets (CM). Using these sequences as input, the program MEME

(*Bailey and Elkan, 1995*) was used to recover a motif logo, requiring candidate motifs to appear in at least 90% of the input sequence set. We found only one motif for each pool of sequences that satisfied this criterion.

## Gene Ontology analysis

The list of SRSF1 translational targets, including those with CLIP-tags and those estimated to be direct translational targets (CLIP-tag and consensus motif) were uploaded as a gene list to the Database for Annotation, Visualization and Integrated Discovery (DAVID) v6.7 (http://david.abcc.ncifcrf.gov/home.jsp), while all the mRNAs detected by RNA-seq were used as a background (*Dennis et al., 2003*). Then we analyzed the over-represented functional categories in 'Biological Process' using the gene functional classification tool containing *all the levels of GO terms* as described in *Huang et al. (2009)*. EASE scores (modified Fisher's exact p value) were computed for all categories. The Benjamini–Hochberg correction method was applied to the data in order to identify the most significantly over-represented gene categories.

## SILAC

For SILAC, HEK 293T cells were grown for 8 d with two passages in DMEM SILAC media before transfection (Dundee Cell Products, Dundee, UK). The arginine and lysine isotopes were as follows: R0K0, L-[$^{12}$C$_6$$^{14}$N4]arginine (R0), and v-[$^{12}$C$_6$$^{14}$N2]lysine (K0); R6K4, L-[$^{13}$C$_6$$^{14}$N4]arginine (R6), and L-[$^{12}$C$_6$$^{2}$H4$^{14}$N2]lysine (K4); and R10K8, L-[$^{13}$C$_6$$^{15}$N4]arginine (R10), and L-[$^{13}$C$_6$$^{15}$N2]lysine (K8). Cells were grown either with R0K0 (untransfected cells), R6K4 (cells transfected with empty vector), or R10K8 (cell transfected with pCG-T7SRSF1). At 48 hr after transfection, cells were washed twice in ice-cold PBS and scraped into ice-cold RIPA buffer containing a protease inhibitor cocktail (Roche). Total protein extracts were measured by the Bradford assay. Equal amounts of protein from unlabeled and labeled samples were run on SDS–PAGE, and gel lanes were cut into 10 sections, followed by overnight digestion with trypsin at 37°C. Sample processing, mass spectrometry, and data analysis were performed by the Dundee Cell Products service.

## Comparison with the SILAC experiment

The protein quantification changes detected by SILAC were compared to the changes in PSR detected with RNA-seq upon SRSF1 overexpression. After applying quantile normalization (*Bolstad et al., 2003*) to the enrichment signal of SRSF1 versus untransfected, and to the enrichment signal of the empty vector sample over untransfected, the log$_2$ ratio of the two normalized signals was considered. In order to compare proteins identified by SILAC with mRNAs shifted to polysomes, the IPI identifiers (*Kersey et al., 2004*) from the SILAC experiment were mapped to the RefSeq identifiers used for the RNA-seq data.

## RNA purification and array analysis

Total RNA was purified and genomic DNA removed using the RNeasy Plus Kit (74,134; Qiagen) according to the manufacturer's instructions. The RNA quality was verified by the 2100 Bioanalyzer (Agilent). Each array experiment was performed in triplicate. Array data analysis was performed by GenoSplice (www.genosplice.com; Paris, France). Affymetrix Human JAY arrays were normalized using the probe scaling method and background corrected with ProbeEffect from GeneBase (*Kapur et al., 2008*). The gene expression index was computed from probes that were selected using ProbeSelect from GeneBase (*Kapur et al., 2008*). Gene expression signals were computed using these probes. Genes were considered expressed if mean intensity was ≥200. Genes were considered regulated if: (1) they were expressed in at least one condition (i.e., SRSF1 and/or empty vector control); (2) fold-change was ≥1.5; and (3) the unpaired *t* test p value between gene intensities was ≤0.05. For each probe, a splicing index was computed. Unpaired *t* tests were performed to test the difference in probe expression between the two samples as described previously (*Shen et al., 2010*). Probe p values in each probeset were then summarized using Fisher's method. Using annotation files, splicing patterns (cassette exons, 5′/3′ alternative splice sites, and mutually exclusive exons) were tested for difference between isoforms, selecting those with a minimum number of regulated probesets (with p≤0.01) in each competing isoform (at least one third of 'exclusion' probesets have to be significant, and at least one third of 'inclusion' probesets have to be significant and show an opposite regulation for the splicing index compared to 'exclusion' probesets). For example, for a single cassette exon, the exclusion junction and at least one of the three inclusion probesets (one exon probeset and two inclusion junction probesets) have to be significant and have to show an opposite regulation for the splicing index.

## Comparison of translational targets and array experiment

Significantly regulated cassette exons from the array were mapped to RefSeq genes. Each upregulated or downregulated exon was assigned to alternative isoforms that included or excluded the exon, respectively. No isoform with two exons regulated in opposite directions. The values of PSR were compared between the datasets of mRNAs either including or skipping the cassette exons from the array, as well as for all mRNAs detected by RNA-seq belonging to genes with multiple isoforms.

## Immunofluorescence, microscope image acquisition, and processing

Appropriately transfected cells were seeded on coverslips in six-well plates. After 24 hr, cells were rinsed in PBS and fixed with 4% paraformaldehyde for 10 min at room temperature. The fixed cells were washed with PBS and permabilised using PBS + 0.2% Triton. After washing, the slides were blocked in 1% BSA in PBS for 1 hr. Various antibodies were used including anti-pericentrin, anti-α-tubulin, and anti-SRSF1, which were diluted 1:2000, 1:1000, and 1:1000, respectively. Slides were then washed and incubated with the appropriate AlexaFluor 488 and AlexaFluor 594 labeled secondary antibodies at 1:2000. Final washes were followed by mounting with DAPI Vectashield. A fluorescent upright microscope, Zeiss Axioplan 2, was used to image the cells at 20× magnification. At least 100 mitotic cells were captured and abnormal mitoses were scored for multipolar spindle and metaphase misalignment. Data are presented as a percentage of abnormal phenotypes versus total number of scored mitotic cells.

## Live cell imaging

Transfected cells were seeded in six-well plates. After 24 hr, cells were rinsed twice in PBS and grown in phenol-red free medium. Live imaging for 24 hr at a time lapse of 15 min was performed with a Zeiss Axiovert 200 microscope. Metamorph software was used for image capture and analysis. At least 20 mitotic cells were assessed for the time spent in mitosis.

## Major datasets

Supplementary files 1–4 are available at Dryad: *Maslon et al. (2014)*.

Supplementary file 1. List of SRSF1 translational targets. List of 1576 mRNAs enriched in polysomes upon SRSF1 overexpression.

Supplementary file 2. SILAC analysis following SRSF1 overexpression. List of proteins identified in SILAC experiment.

Supplementary file 3. Alternative splicing analysis. List of the 382 SRSF1-regulated cassette exons determined by exon arrays (cassette exons). List of SRSF1-regulated cassette exons with SRSF1-clip-tags over cassette exon (clip tag over cassette exon tab).

Supplementary file 4. Bed file containing positions and sequences of the consensus motif in hg18 assembly of human genome.

# Acknowledgements

We thank Carol-Anne Martin for the generous gift of antibodies and constructs and for discussions, Nick Gilbert for HT1080 cells stably expressing GFP-CENPA and for technical assistance with fractionation, Mathew Pearson for help with life cell imaging, and Eneritz Agirre for help with alternative splicing analysis. We are also grateful to Noemi Fernandez Sanchez and Dasa Longman for discussions and advice and to Andrew Wood for critical reading of the manuscript.

# Additional information

## Funding

| Funder | Grant reference number | Author |
| --- | --- | --- |
| MRC Core Funding | | Sara R Heras, Javier F Cáceres |
| Wellcome Trust Senior Investigator Award | Grant 095518/Z/11/Z | Magdalena M Maslon, Javier F Cáceres |
| Marie Curie | IntraEuropean fellowship | Sara R Heras |
| Consolider RNAReg | CSD2009-00080 | Nicolas Bellora, Eduardo Eyras |

| Funder | Grant reference number | Author |
|---|---|---|
| Ministerio de Economia y Competitividad | BIO2011-23920 | Nicolas Bellora, Eduardo Eyras |
| Sandra Ibarra Foundation | FSI2011-35 | Nicolas Bellora, Eduardo Eyras |

The funders had no role in study design, data collection and interpretation, or the decision to submit the work for publication.

### Author contributions

MMM, SRH, Conception and design, Acquisition of data, Analysis and interpretation of data, Drafting or revising the article; NB, EE, Analysis and interpretation of data, Drafting or revising the article; JFC, Conception and design, Analysis and interpretation of data, Drafting or revising the article

## Additional files

### Major datasets

The following dataset was generated:

| Author(s) | Year | Dataset title | Dataset ID and/or URL | Database, license, and accessibility information |
|---|---|---|---|---|
| Maslon MM, Heras SR, Bellora N, Eyras E, and Cáceres JF | 2014 | Data from: the translational landscape of the splicing factor SRSF1 and its role in mitosis. Dryad Digital Repository, 2014 | http://dx.doi.org/10.5061/dryad.5sv39 | Publicly available at the Dryad digital repository. |

The following previously published dataset was used:

| Author(s) | Year | Dataset title | Dataset ID and/or URL | Database, license, and accessibility information |
|---|---|---|---|---|
| Sanford JR, Wang X, Mort M, Vanduyn N, Cooper DN, Mooney SD, Edenberg HK, Liu Y | 2009 | Splicing factor SFRS1 recognizes a functionally diverse landscape of RNA transcripts | http://sanfordlab.mcdb.ucsc.edu/Sanford_Lab/Datasets.html | Freely available online. |

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
