## [Decision Letter]

Thank you for sending your work entitled “The translational landscape of the splicing factor SRSF1 and its role in mitosis” for consideration at *eLife*. Your article has been favorably evaluated by a Senior editor, a Reviewing editor, and 2 reviewers, one of whom, Ivan Topisirovic, has agreed to reveal his identity.

The Reviewing editor and reviewers discussed their comments before reaching this decision, and the Reviewing editor has assembled the following comments to help you prepare a revised submission.

You provide evidence that SRSF1 controls expression of genes encoding cell cycle regulators by orchestrating splicing and translation of corresponding mRNAs. At the physiological level, you show that this function of SRSF1 is required for mitotic progression. Using RNA-Seq you identified mRNAs that become enriched in polysome fractions upon overexpression of the SR protein SRSF1 in 293 cells. A fraction of these RNAs are directly bound by the protein and are therefore potential direct targets of translational regulation by SRSF1, a conclusion also supported by comparative proteomics data. Of interest, a significant fraction of genes whose alternative splicing is regulated by SRSF1 also show translational regulation, suggesting coordinated regulation of the two processes by this shuttling, multifunctional protein. Enrichment in genes involved in spindle/kinetochore formation led you to study the effects of SRSF1 knock down on cell division. Your results, supported by rescue experiments, indicate that SRSF1 is required for mitotic progression and proper spindle formation. This study will be appealing to the broad readership of *eLife*.

While the function of SRSF1 in splicing and translation control was firmly established by previous studies (including key contributions from your lab) and SRSF1 was known to function as an oncogene able to cause cell transformation through the regulation of cell proliferation and apoptosis, your results are important because a) they provide a genome-wide view of translational regulation by SRSF1 and b) offer solid mechanistic insights on specific events of cell division regulated by SRSF1 and on putative gene targets; this information may be relevant for other splicing factors as well, particularly in the light of abundant recent reports of splicing factors frequently mutated in cancer.

Below are comments that you need to address to improve the paper.

1) Since these studies were carried out in asynchronous cell populations, and considering that Stumpf et al. Mol. Cell (2013) have shown that there is a dynamic reprogramming of translation throughout cell cycle, is it possible that the effects of SRSF1 on cell cycle could be a cause of at least some of the observed changes in translation? These effects could be particularly significant for the cohorts of genes whose expression (including translation) is restricted to specific phases of cell cycle. Along these lines, is the expression of SRSF1 constant throughout the cell cycle?

Further comment on the similarities and differences between the effects of SRSF1 knock down reported in the manuscript and the effects on G2 cell cycle arrest, genomic instability and apoptosis observed by Manley and colleagues in chicken DT40 cells (Li et al. Genes & Dev 19: 2705) would be useful.

2) There are several translational suppressors that you show are regulated by SRSF1, such as Paip2 that could mediate its effects on translation. In the past it has also been shown by you and others that SRSF1 can affect signaling pathways that impinge on translational machinery (mTOR and MAPK). Hence it appears that SRSF1 may affect translation via variety of mechanisms and that this could likely comprise different subclasses of SRSF1 mRNA targets. Therefore dissecting these mechanisms and/or commenting and discussing on these issues would (at least in this reviewer's opinion) improve the present manuscript by providing additional insights in the circuitry of the SRSF1 translation regulation network.

3) Further analysis or discussion of whether SRSF1 overexpression leads to phenotypic effects on cell cycle progression which are reciprocal to those caused by SRSF1 depletion (e.g., enhanced metaphase transitions or reduction of multipolar spindle formation). This information may be also relevant to identify key targets which are rate limiting for cell growth.

4) It would be informative to study the distribution of SRSF1 binding sites in the 330 likely direct mRNA targets of translational regulation by SRSF1, including those cases where both alternative splicing and mRNA translation are coordinately controlled. Mutating the purine rich motif in the context of a couple of novel SRSF1 splicing/translation targets is important to show that these are direct effects of SRSF1 on translation, especially because your previous study the effects are mediated not only by the ESE element, but also by hyperphosphorylation of 4E-BPs via PP2A, that have been recently shown to regulate translation of mRNAs that harbor TOP and similar pyrimidine rich motifs by Ruggero and Sabatini's groups.

5) We recommend validation of selected positive and negative hits in each fraction of the sucrose gradient. Using a polysomal vs. sub-polysomal cut-off although necessary cost-wise for the RNAseq part of the study may be misleading in the sense that certain mRNAs could move only by one fraction, whereas others could exhibit more dramatic shifts. These differences would have a strong impact on expression of corresponding proteins.

6) A more detailed Methods section is required. It is not clear how some of the normalizations were carried out. This is important considering emerging issues with large-scale data analysis in particular in studying translational control where log ratios etc. were used (PMID: 21422072; PMID: 21115840; PMID: 23810193).

7) It is pertinent to see whether following SRSF1 siRNA treatment there is a decrease in protein levels for some of the SRSF1 targets suspected of mediating the bipolar spindle formation.

---

## [Author Response]

*1) Since these studies were carried out in asynchronous cell populations, and considering that Stumpf et al. Mol. Cell (2013) have shown that there is a dynamic reprogramming of translation throughout cell cycle, is it possible that the effects of SRSF1 on cell cycle could be a cause of at least some of the observed changes in translation? These effects could be particularly significant for the cohorts of genes whose expression (including translation) is restricted to specific phases of cell cycle*.

Since our studies were carried out for the most part in asynchronous cell populations, it remains possible that effects of SRSF1 on cell cycle progression could be a cause of at least some of the observed changes in translation. To address this, we have performed additional experiments to monitor the effects of overexpression SRSF1 in cell cycle progression and have concluded that the polysomal shift of mRNA targets upon SRSF1 overexpression is primarily linked to SRSF1-mediated translation effects and not to an indirect effect on the cell cycle (Figure 5—figure supplement 1). This is now discussed in the text in the Results section. See also a more detailed response to point 3, below.

*Along these lines, is the expression of SRSF1*
*constant throughout the cell cycle?*

Recently, the Lamond Lab performed a large-scale proteomic screen, which revealed proteome dynamics during cell cycle (Ly et al. eLife. 2014 Jan 1;3:e01630. doi: 10.7554/eLife.01630.PubMed PMID: 24596151). In this work, no significant changes in SRSF1 protein levels were detected throughout the cell cycle, albeit there was a 1.2 fold increase in SRSF1 protein levels from G1 to S phase. This does not rule out that subcellular localization and/or phosphorylation of SRSF1 could be cell-cycle regulated.

*Further comment on the similarities and differences between the effects of SRSF1 knock down reported in the manuscript and the effects on G2 cell cycle arrest, genomic instability and apoptosis observed by Manley and colleagues in chicken DT40 cells (Li et al. Genes & Dev 19: 2705) would be useful*.

We had referred to those papers from the Manley lab already, but we have now expanded this section.

*2) There are several translational suppressors that you show are regulated by SRSF1, such as Paip2 that could mediate its effects on translation. In the past it has also been shown by you and others that SRSF1 can affect signaling pathways that impinge on translational machinery (mTOR and MAPK). Hence it appears that SRSF1 may affect translation via variety of mechanisms and that this could likely comprise different subclasses of SRSF1 mRNA targets. Therefore dissecting these mechanisms and/or commenting and discussing on these issues would (at least in this reviewer's opinion) improve the present manuscript by providing additional insights in the circuitry of the SRSF1 translation regulation network*.

We do agree that SRSF1 can affect mRNA translation via different mechanisms. This is now discussed in the text in the section describing Figure 3 (end of the second paragraph of the Results section entitled “Validation of SRSF1 translational targets).

*3) Further analysis or discussion of whether SRSF1 overexpression leads to phenotypic effects on cell cycle progression which are reciprocal to those caused by SRSF1 depletion (e.g., enhanced metaphase transitions or reduction of multipolar spindle formation). This information may be also relevant to identify key targets which are rate limiting for cell growth*.

In order to address whether some of the changes in mRNA translation induced by SRSF1 could be due to indirect effects of SRSF1 in cell cycle progression, we performed two additional experiments. First, we observed that increased expression of SRSF1 under the conditions used in the polysomal shift analysis, does not grossly affect the cell cycle profile (Figure 5—figure supplement 1), strongly suggesting that the observed increase in polysomal to subpolysomal ratios for these targets is primarily linked to SRSF1-mediated translation and not to an indirect effect on the cell cycle. By contrast, SRSF1 depletion led to major cell cycle aberrations (Figure 5—figure supplement 1), with approximately 50 % of SRSF1-depleted cells remaining arrested in G2/M phase. This suggests that a certain threshold of SRSF1 is required to progress normally through the cell cycle. This could be caused, at least partially, by a loss of SRSF1-dependent translation of targets required for cell cycle progression, as we observed decreased association of these mRNA targets with polysomes in SRSF1-depleted cells (Figure 5), as well as decreased levels of the corresponding proteins (Figure 5).

*4) It would be informative to study the distribution of SRSF1 binding sites in the 330 likely direct mRNA targets of translational regulation by SRSF1, including those cases where both alternative splicing and mRNA translation are coordinately controlled*.

We analyzed the position bias with respect to the SRSF1 consensus motif in the 5’UTR, CDS and 3’UTR regions of mRNA translational targets (New Figure 2—figure supplement 2). In addition, we now also attach a bed file containing information on the position of consensus site mapped to CLIP regions, which allows checking the position and sequence of the translation motif in SRSF1 translational targets (Supplementary file 4).

*Mutating the purine rich motif in the context of a couple of novel SRSF1 splicing/translation targets is important to show that these are direct effects of SRSF1 on translation, especially because your previous study the effects are mediated not only by the ESE element, but also by hperphosphorylation of 4E-BPs via PP2A, that have been recently shown to regulate translation of mRNAs that harbor TOP and similar pyrimidine rich motifs by Ruggero and Sabatini's groups*.

This is a good idea in principle, but in practical terms, mutation of a purine-rich element recognized by SRSF1 can also affect splicing/alternative splicing decisions, complicating the analysis. In fact, this type of approach has been previously used by us with Luciferase reporters, where it is easier to control these effects (See Sanford et al. (1994) Genes Dev.; Michlewski et al. (2008) Mol Cell). In those studies we observed a decreased association of the corresponding mutated mRNA with polysomes. We think that this sufficiently proves that the binding to the target mRNA is required for SRSF1 effect on its target translation.

We have previously shown that SRSF1 promotes translation initiation of bound mRNAs by suppressing the activity of 4E-BP, a competitive inhibitor of cap-dependent translation. This activity is mediated by interactions of SRSF1 with components of the mTOR signaling pathway. However, this was shown with mRNA reporters. In order to determine whether endogenous SRSF1 translational targets responded to the same mechanism of translational activation we treated cells transiently expressing SRSF1 (or control cells) with a specific inhibitor of the mTOR kinase, PP242 (Dowling et al. 2010) and then measured polysomal to subpolysomal ratios of a subset of selected SRSF1 translational targets. Interestingly, we found that inhibition of mTOR abrogated the stimulatory activity of SRSF1 on the translation of selected targets (Figure 3). This demonstrates that the activity of SRSF1 in translational activation of endogenous targets requires the mTOR pathway.

*5) We recommend validation of selected positive and negative hits in each fraction of the sucrose gradient. Using a polysomal vs. sub-polysomal cut-off although necessary cost-wise for the RNAseq part of the study may be misleading in the sense that certain mRNAs could move only by one fraction, whereas others could exhibit more dramatic shifts. These differences would have a strong impact on expression of corresponding proteins*.

In principle, we agree with this statement and approach and we have attempted to carry such analysis for a small subset of SRSF1 translational targets. We obtained results that confirmed the results obtained with the polysomal shift, but do not show clear differences for individual fractions. We believe that a way forward would be to use ribosome profiling to study in more detail how SRSF1 is activating the translation of individual mRNAs. This will be the base for future studies.

*6) A more detailed Methods section is required. It is not clear how some of the normalizations were carried out. This is important considering emerging issues with large-scale data analysis in particular in studying translational control where log ratios etc. were used (PMID: 21422072; PMID: 21115840; PMID: 23810193)*.

Most studies determine mRNAs translation levels as log ratios of actively translated mRNAs divided by the corresponding cytosolic mRNA levels, obtained in parallel. However, recent work from Larsson and colleagues (PMID: 21115840) showed that this method resulted in a significant number of false positives and false negatives. These authors developed a new approach for the analysis of translational activity. Accordingly, we did not use cytosolic mRNA to normalize; instead we calculated the proportion of the abundance of a given mRNA in polysomes, relative to the abundance of this mRNA in both polysome and subpolysomes (Eq. 2 in the manuscript). Subsequently, the log-rates of these ratios were calculated (Figure 1). This is now explained in more detail in the Results section.

*7) It is pertinent to see whether following SRSF1 siRNA treatment there is a decrease in protein levels for some of the SRSF1 targets suspected of mediating the bipolar spindle formation*.

We are now adding a new panel showing protein levels of selected targets involved in bipolar spindle formation following SRSF1 depletion (new Figure 5).